# *Labor Space*: A Unifying Representation of the Labor Market via Large Language Models

## ABSTRACT

The labor market is a complex ecosystem comprising diverse, interconnected entities, such as industries, occupations, skills, and firms. Due to the lack of a systematic method to map these heterogeneous entities together, each entity has been analyzed in isolation or only through pairwise relationships, inhibiting comprehensive understanding of the whole ecosystem. Here, we introduce *Labor Space*, a vector-space embedding of heterogeneous labor market entities, derived through applying a large language model with fine-tuning. Labor Space exposes the complex relational fabric of various labor market constituents, facilitating coherent integrative analysis of industries, occupations, skills, and firms, while retaining type-specific clustering. We demonstrate its unprecedented analytical capacities, including positioning heterogeneous entities on an economic axes, such as 'Manufacturing–Healthcare'. Furthermore, by allowing vector arithmetic of these entities, Labor Space enables the exploration of complex inter-unit relations, and subsequently the estimation of the ramifications of economic shocks on individual units and their ripple effect across the labor market. We posit that Labor Space provides policymakers and business leaders with a comprehensive unifying framework for labor market analysis and simulation, fostering more nuanced and effective strategic decision-making.

## CCS CONCEPTS

• **Applied computing** → **Economics**; • **Computing methodologies** → *Information extraction*; • **Information systems** → *Environment-specific retrieval*.

## KEYWORDS

Labor Market, Word Embedding, Skill, Job, Industry, Firm, Large Language Model

**ACM Reference Format:**
Anonymous Author(s). 2023. *Labor Space*: A Unifying Representation of the Labor Market via Large Language Models. In *Proceedings of The Web Conference (WWW' 24)*. ACM, New York, NY, USA, 11 pages. https://doi.org/XXXXXXX.XXXXXXX

## 1 INTRODUCTION

Understanding the labor market is essential to comprehend the entire economy. It's like a bridge connecting individual skills and

workforce dynamics to the bigger economic picture. A strong understanding of the labor market can not only provide macroeconomic status, such as unemployment rate and household income, but also reveal where our entire economic system moves toward by grasping the complex inter-relationship between various economic units in the economy.

However, analyzing the labor market is a challenging task due to its inherent complexity. It involves various interacting components, from individual skills and jobs to large industries and businesses. Human capital, encompassing the diverse skills and jobs populating the labor market, is the basis upon which industries and firms are built. It represents the nuanced interplay of individual capabilities and the broader economic forces. For instance, the rise of a new technological skill could dictate the trajectory of entire industries, and subsequently the businesses within them. Hence, understanding this connection between human capital and larger economic entities is crucial, not only for academic comprehension but also for pragmatic decision-making in industries and policymaking.

Yet, despite its significance, a holistic framework that encompasses these interconnections has been absent. Most existing methods either focusing only on a specific entity or pairwise relationships, without considering the entire structure where different types of entities are entangled. This gap in analysis often results in oversimplified representations and conclusions, limiting the depth of insights that can be derived about the labor market's ecosystem.

Here, we present Labor Space, an unifying representation of the heterogeneous entities in the labor market. By leveraging the capabilities of large language model — Google's BERT — with additional fine-tuning with representative descriptions of the labor market entities from various corpora, we derive an unifying representation of the labor market's constituents, including skills, occupations, industries, and firms. The landscape of Labor Space provides a holistic framework, by exposing the relational fabric of various labor market constituents. At the same time, it facilitates the clustering of related industries, occupations, skills, and firms, while retaining type-specific clustering.

Also, by systematically mapping heterogeneous units onto sensible semantic axes, such as 'Manufacturing-–Healthcare', Labor Space offers unprecedented analytical capacities. Furthermore, through intricate vector arithmetic of these entities, Labor Space enables the exploration of complex inter-unit relations, thereby allowing estimation of the impact of economic shocks or adoption of new technology, such as artificial intelligence, on individual entities and their ripple effect across the labor market.

Our overarching goal with Labor Space is to bridge the existing analytical gap, connecting the dots between individual skills and jobs and the vast expanse of industries and firms. This work underscores the significance of viewing the labor market not as isolated silos but as an interconnected web, where shifts in human capital can ripple across the entire economic ecosystem. We expect that

**Table 1: Description Data and Source**

| Entity | Data Source | Number of Entities | Example |
|---|---|---|---|
| Industry | NAICS | 308 | Metal ore mining comprises establishments primarily engaged in developing mine sites or mining metallic minerals, and establishments primarily engaged in ore dressing and beneficiating (i.e., preparing) operations, such as crushing, grinding, washing, drying, sintering, concentrating, calcining, and leaching. ~ |
| Occupation | O*NET | 1,016 | Data scientists develop and implement a set of techniques or analytics applications to transform raw data into meaningful information using data-oriented programming languages and visualization software. |
| Skill | ESCO | 307 | Counseling assists others to gain access to social, legal or other services and benefits, including making referrals to other professionals and organizations. |
| Firm | Crunchbase | 489 | Meta is a social technology company that enables people to connect, find communities, and grow businesses. Previously known as Facebook, Mark Zuckerberg announced the company rebrand to Meta on October 28, 2021 at the company's annual Connect Conference. ~ |

our Labor Space empowers stakeholders, from business leaders to policymakers, with a holistic view of the labor market, enabling more informed, strategic, and effective decisions.

## 2 RELATED WORKS

### 2.1 Analysis of the labor market and its entities

The labor market, a multifaceted and intricate ecosystem, is woven together by interconnection between entities like industries, occupations, skills, and firms. Historically, research into this domain has predominantly fallen under two analytical frameworks: (1) understanding the influence of one entity over another, and (2) elucidating the structure of one entity in juxtaposition with another.

The first paradigm largely revolves around the human capital perspective. Human capital in economic theory, denotes the qualitative attributes of labor, extends traditional production factors, such as land, labor, and tangible capital, incorporating a worker's skills and expertise. This perspective delves deep into the impact of investments made in human resources, scaling from individual workers to overarching industries and firms [7, 30]. Existing human capital studies have ventured into areas like knowledge spillovers [3, 39–41], in addition to the private and societal returns linked to industry- and occupation-specific human capital [32, 35, 38].

Network analysis in economics, meanwhile, has illuminated intricate relationships between different entities. For instance, the skill network, inferred from the co-occurrence of skills among job seekers and providers, suggests that individuals with a diverse skill set command higher wages; more remarkably, those synergizing their varied skills earn top-tier wages [5]. Further exploring this, Alabdulkareem et al. outlined a skill network, pinpointing the dichotomy between physical and cognitive skills based on job requirements [2]. Additionally, the ties binding educational paths, as determined by collaboration within organizations, have underscored the wage benefits of working alongside those with complementary qualifications [34]. Beyond skills, the labor market research also delves into the broader associations between firms and industries, helping demystify the macro-structures of the global economy [33, 36].

However, despite the valuable insights these studies offer, there remains an important oversight in their approach. Much of the existing research tends to examine entities in isolation or restricts

itself to binary relationships, whose narrowed focus hinders a holistic grasp of the labor market's nuances. A significant limitation has been the absence of a cohesive methodology capable of simultaneously mapping the varied units of the labor market. This gap in research underscores the urgent need for a more integrated analytical framework that bridges the existing divides.

### 2.2 Application of language models in social science

Language models, such as word embedding, transform high-dimensional or unstructured text data into lower-dimensional representations [28]. In these models, embedding vectors are derived from the relationships of words within sentences, and each vector captures the semantic essence or underlying meaning of its corresponding word. With the introduction of word or document embedding models into the social sciences [29], a new avenue has emerged for analyzing and understanding cultural, social, and historical trends [26].

Through the use of word embeddings in social science, deeper insights into historical, cultural nuances, and societal biases have been achieved. An exploration of the vast corpus of English-language Google Books revealed the evolution and persistence of social group representations over two centuries, using word embeddings from 850 billion words [14].

A deep dive into semantic biases, trained on standard web text, showed that our historical biases, ranging from the benign to the problematic, are deeply embedded in everyday human language [12]. This finding aligns with studies that highlighted the changing dynamics of stereotypes, especially in terms of gender and ethnicity in the U.S. during the 20th and 21st centuries. Notably, these models identified significant societal events, such as the 1960s women's movement and immigration trends [21].

Beyond social stereotypes, the flexibility of word embeddings has been showcased in other domains. Insights into socio-economic classes and their transformations over a century were derived using word embeddings, linking cultural theories with semantic relations in high-dimensional spaces [23]. Using embeddings from Google News articles, the persistent gender biases in occupational terms evident even in modern media were unveiled [10]. In the realm of management science, the importance of language models

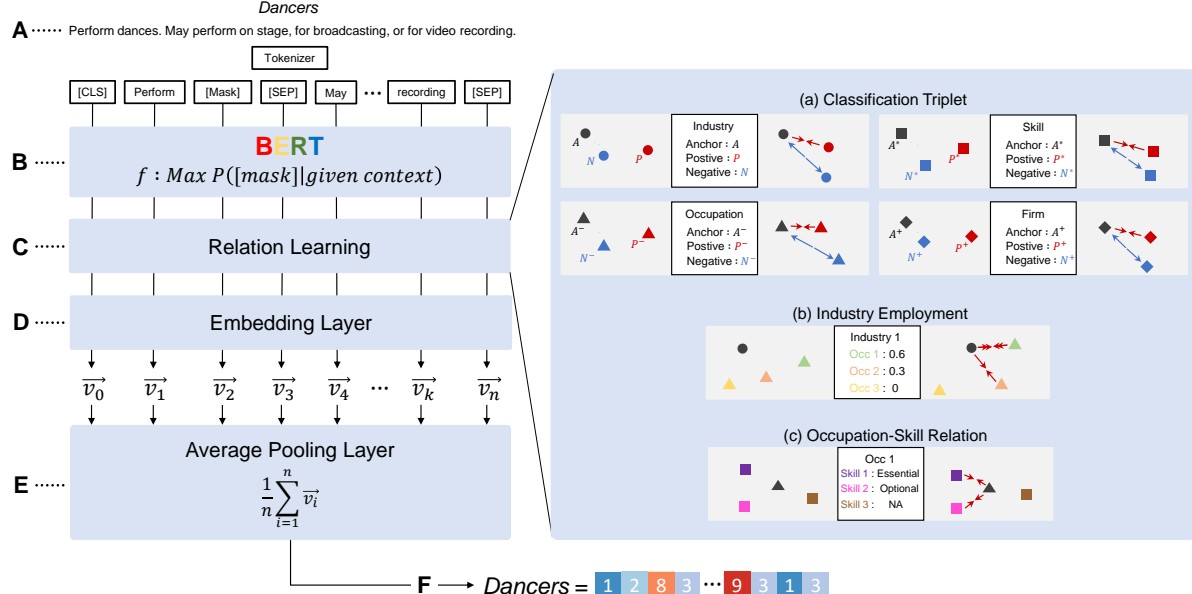

**Figure 1: Constructing the Labor Space. (A) Sample entity description from the 2,120 available. (B) Google's BERT, fine-tuned with descriptions from NAICS, O\*NET, ESCO, and Crunchbase, predicts the [Mask] token using its context, learning labor market nuances. (C) We craft inter-relations between Labor Space entities using paired datasets, as magnified in the right-side figure. (D) Both contextual and relational information is captured in BERT's final hidden layer, from which we extract word vectors. (E) A full description vector is represented by averaging its word vectors. (F) Each vector is then labeled with its corresponding title. This results in a vectorized representation of diverse labor market units, illustrating their inter-relations and trajectories in the vector space.**

in business analytics was emphasized. A 'culture dictionary' developed using a word embedding model trained on earnings call transcripts connected corporate culture with tangible business outcomes, expanding our perspective on corporate innovation beyond just factors like R&D spending [25].

However, while these aforementioned studies have been monumental in their revelations and have extensively advanced our understanding of social dynamics using word embeddings, they predominantly emphasize individual layers of units and their associated relationships. In this study, we aims to offer a more interconnected view, integrating various elements of the labor market into a cohesive space. By navigating the relationships between diverse entities, we propose a novel framework to utilize word embeddings in social science research, especially in the context of the labor market.

## 3 DATA AND METHODS

### 3.1 Descriptions

*3.1.1 NAICS.* We use the North American Industry Classification System (NAICS) to embed industry entities into the Labor Space. You can find an example of an industry description in Table 1. NAICS is the standard classification system used by federal statistical agencies in the United States to classify business establishments. It maps business activities onto a hierarchy of industry classifications ranging from 2-digit to 6-digit, depending on the scope and range of the activity. In this study, we focus on the 4-digit industry classification, which includes 308 distinct titles and descriptions.

*3.1.2 O\*NET.* The occupation information is derived from the Occupational Information Network (O\*NET). The O\*NET data comprehensively describes various professions in the contemporary American workplace and is widely used in academic research. Table 1 provides an example of an occupation description for 'data scientist'. Our analysis focuses on 1,016 distinct occupation titles and descriptions from the O\*NET 27.3 database.

*3.1.3 ESCO.* The skill components are derived from European Skills, Competences, Qualifications, and Occupations (ESCO), a multilingual classification system for the European workforce. It offers roughly 15,000 skill units and a hierarchy system ranging from level 0 to level 3. We selected a level 3 skill hierarchy encompassing 307 distinct skill names and descriptions for our analysis. An example description is presented in Table 1.

*3.1.4 Crunchbase.* The firm entities are sourced from *Crunchbase.com*, a platform that provides information on companies, investors, and industry trends. Here, as a representative sample of firms, we selected the firms listed as the S&P 500 companies in the U.S. stock market and extracted their descriptions. As an example, the description of a social media company 'Meta' is presented in Table 1.

## 3.2 BERT model

To quantify the conceptual similarities among heterogeneous types of entities in the labor market, we use the widely-adopted pre-trained word embedding model, Bidirectional Encoder Representations from Transformers (BERT) [16]. Studies have shown that embedding models are capable of representing rich semantic relationships between words through spatial relationships in a vector space [4, 17, 28, 29, 31]. BERT, as an encoder model, is the de-facto standard for contextual representation model. It has been widely employed, especially with fine-tuning, to achieve breakthrough performance in various natural language processing tasks [16].

## 3.3 Fine-tuning for context learning

While the base BERT model is considered reliable for general context, existing studies have reported the relatively lower performance of the model capturing the micro-relationships of entities focusing on a specific context, such as scientific or medical text [8, 13, 24]. Hence, we fine-tune the original BERT model in two ways to capture the latent structure of the labor market. First, we use *HuggingFace*'s "fill mask" pipeline for context learning. Here, the context learning aims to adjust the pre-trained model to the context of the labor market, through additional training with a domain-specific corpus for each entity. As a domain-specific corpus, we concatenate (1) 308 NAICS 4-digit descriptions, (2) O*NET's descriptions for 36 skills, 25 knowledge domains, 46 abilities, 1,016 occupations, (3) ESCO's descriptions for 15,000 skills, 3,000 occupations, and (4) 489 Crunchbase S&P 500 firm descriptions, excluding their labels. We set the maximum token length to 512 and configured the hyperparameters for three epochs, using a batch size of 8 and a learning rate of 2e-5 on an RTX 3080 Ti GPU.

## 3.4 Fine-tuning for relation learning

After the initial fine-tuning to embed the context of the labor market, we conducted an additional fine-tuning process to incorporate inter-entity relatedness. Inspired by the recent work by Cohan et al. [15], which builds interconnections between scientific papers using citation networks, we constructed the following three datasets to train the connections between different types of labor market entities: (1) classification triplet examples, (2) industry-occupation pairs, and (3) occupation-skill pairs.

First, the classification triplet consists of three items — an anchor, a positive sample, and a negative sample. The anchor is the target item that our model aims to learn the relational representation for, while the positive and negative examples are items related and unrelated to the target, respectively. We randomly assign the anchor from 308 industries, 1,016 occupations, 307 skills, and 489 firm descriptions. Positive and negative items are then assigned by leveraging the classification hierarchy system. An entity description is assigned to the positive sample if they share the same parent in their classification system, and to the negative sample if they do not. For each type of entity, we use the following corresponding classification systems: 2-digit NAICS classes for industries, 2-digit SOC's classes for occupations, second-level ESCO classes for skills, and 2-digit General Industry Classification System (GICS) classes for firms. We use triplet loss as the loss function in triplet-based models. It encourages the anchor embedding to be closer to the positive and

farther from the negative, improving the model's discriminative ability in the embedding space (Fig. 1C(a)).

Second, we employ the Occupational Employment and Wage Statistics (OEWS) to build connections between industries and occupations. The OEWS provides the number of workers across occupations within each industry. By calculating the proportion of employment for each occupation in an industry, we identify which occupations are most strongly associated with a given industry. With this relational data, we define the relatedness between industries and occupations using a cosine similarity loss function for training our model. This method encourages the model to produce representations that are more similar for industries with a higher proportion of shared occupations (Fig. 1C(b))

Lastly, to train relations between occupations and skills, we utilize the ESCO dataset, which labels skills as essential, optional, or irrelevant for each occupation. From this information, we construct anchor and positive sample pairs, setting occupation descriptions as anchor samples and skill descriptions as positive samples when the relationship between an occupation and a skill is either essential or optional. We employ the multiple-negatives-ranking loss function to establish the connection between occupations and skills. This loss function uses occupation and skill relational data to adjust the weights so that they are proximal in the vector space (Fig. 1C(c))

## 3.5 Obtaining vectors for labor market entities

To obtain vectors for labor market entities, we first process the textual descriptions of the entities, using the BERT's tokenizer function. The BERT tokenizer, known as Wordpiece, encodes raw text data into token sequences and maps these tokens to their respective token IDs. This conversion turns the description into token sequences, which are the smallest semantic units BERT can interpret. BERT then maps these token sequences to a matrix, where each row comprises 768-dimensional vectors representing each token ID (Fig. 1D). To derive a singular representation of the input description, we compute a linear combination of individual word vectors. This is achieved by summing the embeddings of all the words in the sequence and dividing by the word count (Fig. 1E), thus capturing the overall semantic essence of the description. Each description embedding is labeled with its corresponding title for identification.

## 4 LANDSCAPE OF *LABOR SPACE*

Labor Space provides an embedding environment in which industries, occupations, skills, and firms are plotted in a unifying vector space. The current version includes 308 industries, 1,016 occupations, 307 skills, and 489 firms. However, more types and entities can be integrated when their representative description texts become available. The proximity of entity vectors in our Labor Space, measured via cosine similarity, aptly mirrors their conceptual similarity in the labor market.

Fig. 2A provides a visual representation of the Labor Space in two dimensions by UMAP algorithm [27], where colors distinguish entity types. Labor market entities align with their conceptual resemblances in the labor market. S&P 500 firms predominantly cluster around specific industries, while occupations and skills bridge the spatial gap, reflecting the diverse skill sets and roles required

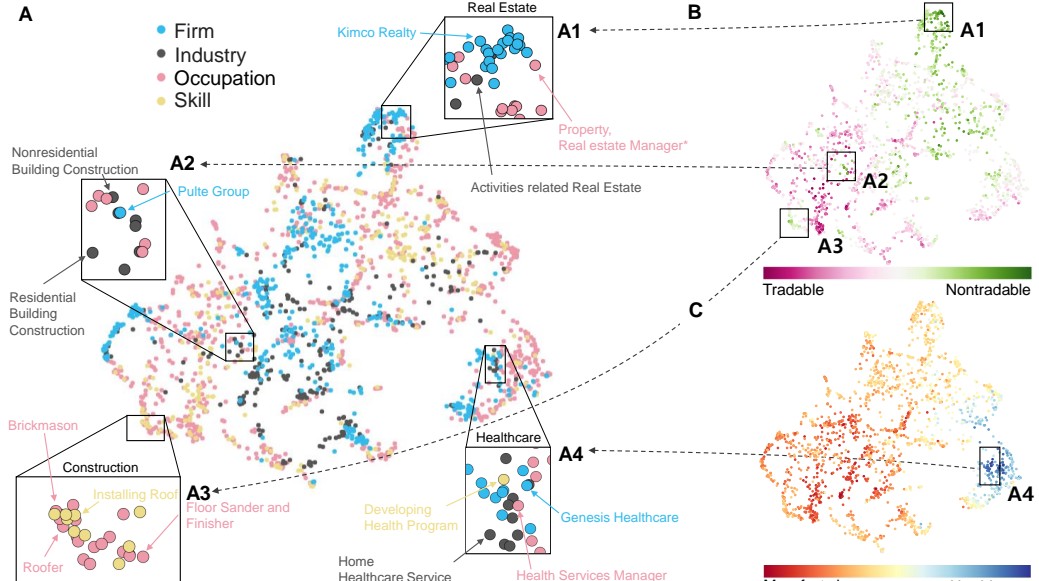

**Figure 2: Visualizing Labor Space. (A) Labor entities, originally 768-dimensional, are mapped to a 2D space using UMAP. (A1) Highlighted values in the Tradable–Nontradable dimension show close ties with real estate. (A2, A3) Construction-related entities cluster due to the industry's blend of manufacturing and tradability. (A4) Emphasized values on the Manufacturing–Healthcare dimension show deep ties to healthcare. (B) Map colored by cosine similarity between V(Tradable → Nontradable) and labor vectors; black rectangles indicate locations from A1, A2, A3. (C) Distribution of cosine similarity between V(Manufacturing → Healthcare) and labor vectors; the black rectangle pinpoints the location in A4.**

by large corporations, especially those in the S&P 500. Moreover, diverse labor market entities cluster spatially based on conceptual similarities. For example, entities associated with real estate (Fig. 2A1), construction (Fig. 2A2 and Fig. 2A3), and healthcare (Fig. 2A4) are grouped closely, while there exists a spatial separation based on specific tasks, such as the one between the entities for exterior and interior construction (Fig. 2A2 and Fig. 2A3, respectively). This organized clustering in the Labor Space empowers policymakers and business owners to pinpoint and prioritize the pivotal skills and occupations relevant to a particular industry or company.

## 5 MAPPING HETEROGENEOUS UNITS ON A CONCEPTUAL AXIS

Using vector arithmetic among its entities, Labor Space offers the capability to map various entities across multiple economic dimensions. Fig. 2B and C depict the relative scores of labor market entities on two distinct axes: (1) the Tradable–Nontradable axis, where nontradable industries encompass local services like restaurants, grocery stores, and salons, and tradable industries comprise businesses that produce exportable or importable products [22]; and (2) the Manufacturing–Healthcare axis, which exposes the relative similarities between manufacturing and healthcare & service industries.

To structure an axis, we first identify a representative entity for each pole. Then, we determine a conceptual vector transitioning

from one pole to the other through vector subtraction [37]. Projecting labor market entities onto this axis vector using cosine similarity calculations lets us measure the shared association between the two vectors in a continuous representation [23]. This methodology facilitates visualizing and quantifying how labor entities are positioned along the axis, such as the Manufacturing–Healthcare dimension, within Labor Space.

Fig. 2B illustrates the Tradable–Nontradable dimension superimposed on Labor Space. As the tradable and nontradable categories are not explicitly distinguished among our entities, we introduce an auxiliary process to compute the industry centroids (see Appendix A.1). Entities aligned with the nontradable sector, such as real estate (Fig. 2A1), gravitate towards the top regions of the Labor Space. Conversely, tradable sectors like manufacturing and energy predominantly settle at the bottom.

In a similar manner, Fig. 2C displays the distribution of projection values on the Manufacturing–Healthcare axis throughout Labor Space. Entities linked to manufacturing are mostly situated on the left, whereas those tied to healthcare lean towards the right in the space. A clear transition is noticeable from manufacturing to healthcare as we move from left to right.

These projection maps, centered on economic axes, grant a comprehensive perspective of the labor market structure, reinforcing that our embedding space authentically captures labor market dynamics. To further underscore this analytical effectiveness of Labor Space, Fig. 3 portrays the continuous spectrum produced by the projection of labor market entities along the Manufacturing–Healthcare dimension. Representative entity titles are annotated

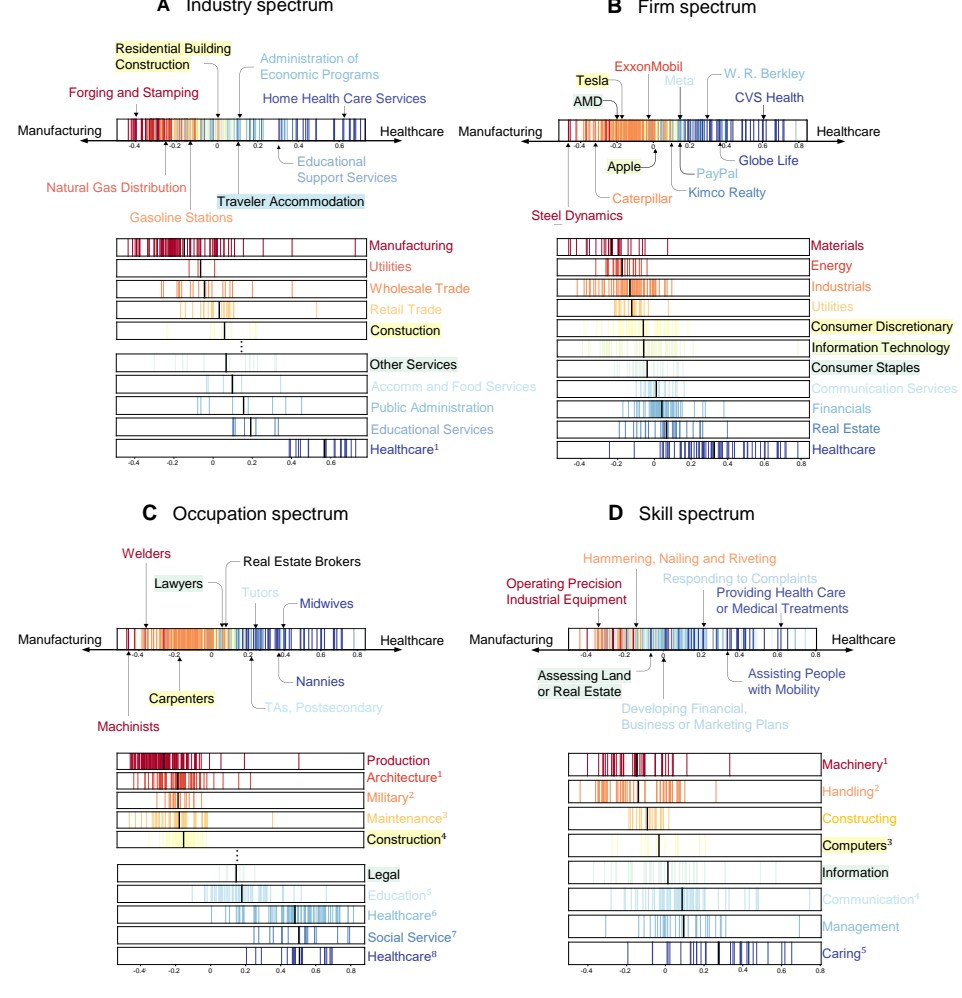

**Figure 3: Spectrum Plot of Labor Market Units.** All labor entities are projected onto the V(Manufacturing → Healthcare) axis. **(A)** Vertical lines within the industry spectrum box show industry embedding projections. Representative industry titles are annotated, using NAICS 2-digit classification for sub-spectrum plotting. **(B-D)** The same projection method applies for firms (using General Industry Classification System), occupations (using Standard Occupation Classification), and skills (using ESCO skill level two hierarchy). The black line represents the mean projection for each class. A consistent alignment of entities along the Manufacturing–Healthcare dimension, underscoring the Labor Space's ability to capture the general alignment across different types of entities.

on this spectrum to spotlight their positions. To validate entity alignment along this axis, we present sub-spectrum plots for each classification system. For industries and occupations, only the top 5 and bottom 5 sub-spectrum plots, sorted by mean projection value for each classification, are accentuated.

Across all labor market entity categories, projecting entities onto the conceptual axis consistently yields reliable outcomes. Firms linked to materials, energy, and industrial utilities (e.g., Steel Dynamics and Caterpillar) are proximate to the manufacturing pole, while those offering services like healthcare, real estate, and finance (e.g., CVS Health and Paypal) are nearer to the healthcare

pole (Fig. 3B). Likewise, skills and occupations tied to manufacturing lean to the left (e.g., Machinists and Welders), whereas those connected to services shift to the right (e.g., Midwives and Nannies) (Fig. 3C and D). This validation underscores the robustness of Labor Space in mapping concepts across diverse economic categories.

## 6 VECTOR ARITHMETIC FOR ECONOMIC ANALOGY

Is it possible to conduct vector arithmetic across types of economic entities? For instance, consider the computation V(Firm A) - V(Skill X) + V(Skill Y). Can such an equation estimate the impact of a new

**Figure 4: Vector analogy of firm and industry entities.**

entity's emergence or the absence of an existing entity from one category on an entity in another category?

Within our Labor Space, vector analogies are employed to uncover latent connections between entities across categories. Fig. 4 presents relationships drawn between firms and industries. In this visualization, the formula V(Firm A) - V(Industry B) + V(Industry C) ∼ V(Firm D) articulates analogical relationships between firms and their corresponding industries. For instance, leading firms in the beverage and restaurant sectors are analogously seen as Nike in the footwear realm, as illustrated in Fig. 4A, B, and C. Another example from Fig. 4D posits that Amazon, if divested of its web search and IT components but equipped with physical stores, would approximate Walmart within the S&P 500 — a deduction made from the vector equation V('Amazon') - V('Web Search Portals, Libraries, Archives, and Other Information Services') + V('Department Stores') ∼ V('Walmart'). Similarly, Tesla, when stripped of its electrical base but supplied with gasoline elements, aligns closely with Ford, as per the equation V('Tesla') - V('Other Electrical Equipment and Component Manufacturing') + V('Gasoline Stations') ∼ V('Ford'), reinforcing our intuitive understanding (see Fig. 4E).

Such vector operations encapsulate a myriad of interactions among labor market entities in reality through vector arithmetics across heterogeneous labor market entities. For instance, the equation 'occupation A + industry B + skill C ∼ firm D', arguably the most intricate analogy in the labor market, exemplifies how vector analogies can be pragmatically harnessed for career recommendations to job aspirants. In our analysis, V('Mathematicians') + V('Other Investment Pools and Funds') + V('Providing Financial Advice') suggests Principal Financial Group, JP Morgan & Chase, and Goldman Sachs, which have been considered as the best firms for mathematicians eager to serve in investment funds, leveraging their financial abilities. Additional examples are presented in Table 3 in Appendix.

## 7 ESTIMATING THE IMPACT OF AI

The labor market is currently experiencing a significant shift due to the widespread integration of artificial intelligence (AI) across numerous economic sectors. AI, broadly defined as technology that discerns patterns from data, has reignited apprehensions regarding technological unemployment [6, 9, 19]. Recent concerns about

AI's influence on the labor market have spurred efforts to gauge occupation-level AI exposure [1, 11, 18, 20], aiming to guide both academic research and policy-making towards facilitating workers' adaptation to the evolving job landscape. Then, does our Labor Space offer a way to gauge AI's footprint on the labor market, spanning across types?

A standout feature of Labor Space is its inherent scalability. Being a vector space shaped by a language model, it can seamlessly admit any new entities, when there are sufficient descriptive texts at hand. To discover whether Labor Space can gauge the labor market reflecting emerging technologies, specifically AI, we compare our results with a prior study quantifying AI industry exposure (AIIE) and AI occupation exposure (AIOE) [18]. Using the top ten AI application definitions from [18] (see Appendix A.2), we derive estimates for AI's imprint on industries and occupations and then correlate these with the aforementioned AIIE and AIOE scores. We collect descriptions of the top ten AI applications, obtain their average embedding, and compute its cosine similarity with both industry and occupation vectors.

Fig. 5A and B illustrate the correlation between our derived cosine similarity scores for AI applications and the established AIIE and AIOE metrics. The X-axis shows exposure scores from [18], assessing each entity's vulnerability to specific AI applications, while the Y-axis presents the cosine similarity between AI application vectors and corresponding Labor Space entries. These metrics exhibit a robust correlation, affirmed by a Pearson's correlation coefficient of 0.51 (p-value < 0.001), suggesting that such cosine similarity measures with novel technology vectors can effectively gauge AI exposure across labor market facets.

Delving deeper, can refining our definitions yield even sharper insights within Labor Space? Leveraging AIIE and AIOE metrics focused on language modeling applications from [18], we isolate descriptions pertinent to language modeling, derive corresponding vectors, and recalibrate our cosine similarity analyses. The correlation between language modeling exposure scores (X-axis) and our computed cosine similarities (Y-axis) notably strengthens for occupational exposure (rising from 0.47 to 0.59) upon this specification, even as industrial exposure maintains its robust correlation of 0.51 (Fig. 5C and D).

Labor Space, given its methodology, tends to underscore heightened AI exposure risks for entities whose tasks align with capabilities of contemporary AI models. For instance, entities like 'Software Publisher' and 'Foreign Language Teachers' are perceived as more vulnerable, while financial domains register diminished AI exposure, appeared in language modeling AI exposure visualizations as well. While determining the absolute accuracy of these estimations remains a future endeavor, analyzing AI exposure through the prism of Labor Space not only underscores its versatility but also furnishes a virtual arena for stakeholders — policymakers, researchers, or business magnates — to conceptualize and simulate potential shifts impacting diverse labor market entities.

## 8 DISCUSSION

The intricate nature of labor markets, characterized by a vast array of interwoven elements ranging from individual skills to expansive industries, has historically posed challenges for comprehensive

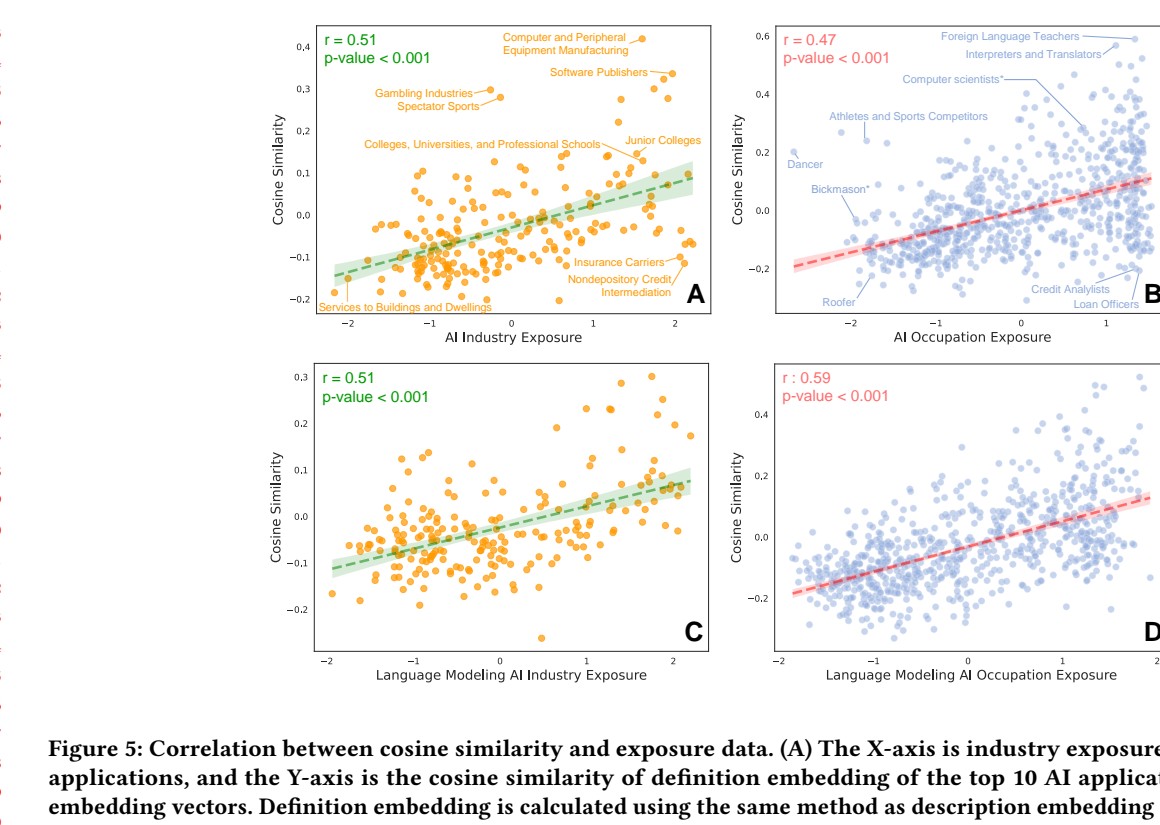

**Figure 5: Correlation between cosine similarity and exposure data. (A) The X-axis is industry exposure data to the top 10 AI applications, and the Y-axis is the cosine similarity of definition embedding of the top 10 AI application list and industry embedding vectors. Definition embedding is calculated using the same method as description embedding of the Labor Space. (B) This figure replaces AIIE to AIOE data. (C) We pick up language modeling among the top 10 AI applications. The X-axis is the industry exposure score of language modeling, and the Y-axis is the cosine similarity between language modeling definition embedding and industry vectors. (D) Correlation plot of language modeling exposure in the occupation level.**

analysis. Despite numerous studies delving deep into specific facets like individual skills or industry trends, a holistic view connecting micro-level human capital to overarching economic entities has remained largely uncharted.

In this study, we introduce *Labor Space*, a pioneering representation that captures the multifaceted entities within the labor market. Utilizing Google's BERT, a state-of-the-art language model, complemented with further refinement, we have succeeded in crafting an unifying portrayal of the labor market's key elements, encompassing skills, occupations, industries, and firms.

The potential applications of Labor Space are vast. For academic researchers, it provides a comprehensive framework that could lead to more nuanced hypotheses and research questions, considering the interconnectedness of labor market entities. On the practical side, policymakers can leverage this to understand the ripple effects of economic or educational policies on various facets of the labor market. Similarly, industry leaders can harness its insights to make informed decisions about skill development, hiring, and industry partnerships.

Still, there are limitations to consider. Firstly, our reliance on Google's BERT, while powerful, ties our results to the biases and constraints of this model. Although BERT is trained on a vast corpus, it may not capture recent developments in the labor market or might reflect societal biases present in its training data. As the labor

market evolves, there's a need for continuous fine-tuning to ensure the relevancy and accuracy of Labor Space.

Another limitation lies in the granularity of data. While Labor Space can cluster related entities, the quality of these clusters largely depends on the input data. Incomplete or outdated data might lead to less accurate representations. Furthermore, while the high-dimensional nature of Labor Space allows for a detailed representation, it also poses challenges in visualization and interpretation. Simplifying this for broader audiences, without losing the depth of information, is an area that requires further exploration.

In addition, while Labor Space captures the interconnectedness of various labor market entities, it may not account for external socio-economic factors or global events that can drastically influence the labor market's dynamics. Future iterations could benefit from integrating external datasets or indicators to provide a more holistic view.

In conclusion, Labor Space presents a groundbreaking approach to understanding the labor market as an ecosystem. While it has its limitations, the potential benefits it offers to both the academic and practical realms are immense. As a future direction, refining the methodology and addressing its constraints will be paramount to ensure its continued relevance and efficacy.

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

**Table 2: Tradability score**

| Percent of Industry | Nontradable | Tradable |
|---|---|---|
| Accommodation and food services | 100 | 0 |
| Administrative and waste services | 89.8 | 10.2 |
| Agriculture, forestry, fishing, and hunting | 0 | 100 |
| Arts, entertainment, recreation | 90 | 10 |
| Construction | 100 | 0 |
| Educational services | 98.89 | 1.11 |
| Finance and insurance | 32.05 | 67.95 |
| Government | 90 | 10 |
| Healthcare and social assistance | 97.8 | 2.2 |
| Information | 34.1 | 65.9 |
| Manufacturing | 0 | 100 |
| Mining 1 | 0 | 100 |
| Other services | 100 | 0 |
| Professional Services | 39.2 | 60.8 |
| Real estate and rental and leasing | 100 | 0 |
| Retail trade | 85.185 | 14.815 |
| Transportation and warehousing | 0 | 100 |
| Utilities | 40 | 60 |
| Wholesale trade | 100 | 0 |

Since the "Mining" industry does not align precisely with the current NAICS 2-digit classification, we employ the "Mining, Quarrying, and Oil and Gas Extraction" to encompass it.

## A APPENDIX

### A.1 Tradability of industry

To make a tradable-nontradable dimension, we set industry centroid with reference to tradability score [22]. Table 2 displays the tradability score for each NAICS 2-digit classification. We designate industries with a score of 100 percent as either tradable or nontradable industry poles.

### A.2 The top 10 AI applications

The Electronic Frontier Foundation (EFF), a respected digital rights nonprofit, has a substantial presence in the academic and research community and collects AI progress statistics from verified sources, including academic literature, blogs, and websites. The EFF selected the top 10 AI applications with recorded scientific progress since 2010, as these are deemed to be experiencing rapid growth and have medium-term relevance. Table 4 gives the top 10 applications list and brief definitions.

**Table 3: Vector analogy of heterogeneous labor market units**

| | Formula | Top 3 entities |
|---|---|---|
| Occupation - Occupation ~ Occupation | V("Data Scientist") - V("Statistician") | 1. Data Architects
2. Database Administrators
3. Data Warehousing Specialists |
| Occupation - Occupation ~ Skill | V("Teller") - V("Cashier") | 1. Monitoring financial and economic resources and activities
2. Managing budgets or finance
3. Analysing financial and economic data |
| Occupation + Industry + Skill ~ Firm | V("Mathematicians")
+ V("Other Investment Pools and Funds")
+ V("Providing Financial Advice") | 1. Principal Financial Group
2. JP Morgan Chase
3. Goldman Sachs |

**Table 4: Top 10 AI application**

| AI application | Definition |
|---|---|
| Abstract strategy games | The ability to play abstract games involving sometimes complex strategy and reasoning ability, such as chess, go, or checkers, at a high level. |
| Image recognition | The determination of what objects are present in a still image. |
| Visual question answering | The recognition of events, relationships, and context from a still image. |
| Image generation | The creation of complex images. |
| Reading comprehension | The ability to answer simple reasoning questions based on an understanding of text. |
| Language modeling | The ability to model, predict, or mimic human language. |
| Translation | The translation of words or text from one language into another. |
| Speech recognition | The recognition of spoken language into text. |
| Instrumental track recognition | The recognition of instrumental musical tracks. |
| Real-time video games | The ability to play a variety of real-time video games of increasing complexity at a high level. |

