# OpenReview forum: "Labor Space: A Unifying Representation of the Labor Market via Large Language Models"
_ACM.org/TheWebConf/2024/Conference — TheWebConf24_

### Official Review · Reviewer_PZKs · 2023-10-28

**Novelty:** 4
**Technical Quality:** 5

**Review:**

This paper presents a representation learning and analytical framework of the labor market, called Labor Space. Labor Space is designed with BERT, a pre-trained language model. The authors first fine-tune BERT with the textual data collected that are related to labor market entities with masked language modeling, and then further fine tune the model with relational training objectives. e.g. triplet loss, industry-occupation relation and skill-occupation relation. Finally, the authors show that the proposed Labor Space can be used for various analytical tasks, such as visualization, visualization along a particular theme, vector analogy, and statistical analysis.

Overall, I do not feel that I have the expertise to properly evaluate this paper, as I am more focused on algorithmic and technical domains, rather than their economic impacts. Given my understanding of this paper, I would like to make the following general comments:

# Strengths:
1. This paper is well-written and easy to follow. I can understand the main idea and arguments of this paper easily.
2. I think this paper brings some insights towards how to quantitatively analyze the labor market, such as visualization (clustering or along an axis), relational analogy, and statistical analysis. These insights may be beneficial to sites such as Linkedin, governmental policy makers, or students who want to choose their major.
3. The techniques used, although not highly novel, are sound and make sense in this context.

# Weaknesses:
1. Overall, from the technical aspect, this paper applies fine-tuned BERT to perform knowledge graph representation and reasoning (e.g. skills, occupations, industry as entities, and the skill-occupation, industry-occupation relations as edges), which from my perspective, is not technically novel, as there are many related works on this (Yao et al, 2019, Zhang et al. 2020). Of course, I am not saying that this paper is not making enough contributions --- the insights are still good.
2. I think the authors can make a better job in showcasing the vector analogy, e.g. by showing some other information, such as the vector distance of the analogy, kNN retrieved entities, etc. This would make the evaluation more convincing.

Yao et al. KG-BERT: BERT for Knowledge Graph Completion, arxiv preprint 1909.03193

Zhang et al. 2020. Pretrain-KGE: Learning Knowledge Representation from Pretrained Language Model. EMNLP 2020

**Questions:**

Q1: See W2. Are there any more information that can be showed to evaluate the vector analogy, e.g. distances, kNN retrieved entities?

Q2: In Figure 2, the authors use manufacturing and healthcare as two poles of the axis, which looks strange to me. It seems that the reverse of manufacturing should be general 'services', instead of healthcare.

**Ethics Review Description:**

Not required

**Reviewer Confidence:**

2: The reviewer is willing to defend the evaluation, but it is likely that the reviewer did not understand parts of the paper

**Scope:**

4: The work is relevant to the Web and to the track, and is of broad interest to the community

---

### Official Review · Reviewer_Rtmd · 2023-11-19

**Novelty:** 5
**Technical Quality:** 5

**Review:**

The paper proposes a model to embed labor text descriptions from multiple sources using finetuning on BERT with loss functions derived from some related heuristics. Then, the paper proposes an analysis on the embeddings and explores some applications (e.g. visualization, grouping, arithmetic, and correlations with metrics from management science).

Strength:
+ Interesting use of LLM on a domain specific application (i.e. labor market)
+ The workflow (finetuning, embedding, visual exploration) is easy to follow and straightforward.
+ Writing is clear.

Weakness:
- Some findings are very straightforward and could be derived by intuition. e.g. The embedding is essentially the last layer of a deep learning model, thus it is mostly linear (and group instances by their classes) according to the loss functions. That explains Figure 2 and 3 easily.

- Lack of in-depth experiments and discussions: the arithmetic is very interesting, as well as the correlations with AIIE and AIOE. However, there are not enough experiments to prove the arithmetic instead of just some examples. Also, those compared metrics are already built on crowdsourced data. Why should we train on text instead? Can't we train a model based on the crowdsourced data to perform the scoring? That is not evaluated or explained.

**Questions:**

I would like to hear the responses from the authors according to the weakness points I raised above.

**Reviewer Confidence:**

3: The reviewer is confident but not certain that the evaluation is correct

**Scope:**

4: The work is relevant to the Web and to the track, and is of broad interest to the community

---

### Official Review · Reviewer_f68m · 2023-11-23

**Novelty:** 3
**Technical Quality:** 3

**Review:**

The authors introduce Labor Space, a unified representation of diverse elements in the labor market. Utilizing Google's BERT and refining it with descriptions from various sources, they create a unified depiction of the labor market, encompassing skills, occupations, industries, and firms. Labor Space reveals the interconnected nature of these constituents, allowing for both broad understanding and specific clustering. The system employs semantic axes for nuanced analysis, such as 'Manufacturing-Healthcare,' enabling deep exploration of inter-unit relations. The authors aim to bridge analytical gaps, emphasizing the interdependence of skills, jobs, industries, and firms. They anticipate that Labor Space will empower stakeholders, from business leaders to policymakers, facilitating more informed and strategic decision-making in response to economic shifts or technological advancements.

**Inadequate Representation of the Labor Market:** The work concentrates on industry, occupation, skill, and firm as the primary components of the labor market, potentially overlooking other crucial elements like wages and external economic forces. This limited focus raises concerns about the work's ability to comprehensively explain the complex labor market ecosystem, as important relationships might be overlooked without considering these additional factors.

**Limited Technical Contribution:** The primary technical contribution of the paper lies in fine-tuning BERT to create a unified representation of the labor market. However, from a technical perspective, this contribution appears limited in scope and may not significantly advance the field.

**Scaled Down Data Sources:** The data sources utilized, including NAICS, O*Net, Esco, and Crunchbase, seem insufficient in terms of both the training data scale and the labor market landscape coverage. The selection criteria, such as focusing only on S&P 500 companies, further narrow the scope of the research, potentially limiting its applicability and generalizability.

**Potential Exposure of Datasets During Fine-Tuning:** It remains uncertain whether BERT has encountered the data sources used in this study before fine-tuning. Given that the data sources are common and publicly available, there is a possibility that the BERT model has already been exposed to this data. This raises questions about the necessity and effectiveness of the fine-tuning process, especially in the absence of validation procedures.

**Missing Empirical Validation:** While the authors conducted some EDA and visualization to show the utility of the embeddings, there is no formal empirical evaluation. The value of the derived embeddings is unclear in the absence of empirical validation, say in a prediction task. Without comprehensive evaluation, I cannot take the embeddings at their face value.

**Questions:**

I am wondering if the authors have conducted any empirical evaluation of the embeddings beyond the EDA.

**Reviewer Confidence:**

4: The reviewer is certain that the evaluation is correct and very familiar with the relevant literature

**Scope:**

3: The work is somewhat relevant to the Web and to the track, and is of narrow interest to a sub-community

---

### Official Review · Reviewer_7CDm · 2023-11-25

**Novelty:** 6
**Technical Quality:** 4

**Review:**

Overall, this paper exhibits a high level of quality. The developed "Labor Space" model is both innovative and pragmatic, underpinned by a logical and well-structured approach. The language and presentation of textual content and visuals are engaging, effectively conveying the core ideas. Furthermore, the limited use of pre-trained models in this field suggests the author's originality. The visualizations and explanations within the paper offer substantial evidence to support the correctness of the work.

This work makes a significant contribution by introducing "Labor Space," which helps uncover intricate relationships among various labor market entities, such as industries, occupations, skills, and firms. It allows for comprehensive analysis while maintaining separate clusters for different entity types. The model also provides unique analytical capabilities, including positioning entities on economic axes and enabling vector arithmetic for exploring inter-entity relations and estimating the impact of economic shocks. "Labor Space" is poised to offer policymakers and business leaders a unified framework for more nuanced and effective labor market analysis and strategic decision-making.

Pros:
1. The novelty of this paper lies in the cross-integration of pre-trained models with the labor market, providing artificial intelligence support to the field of social science, which is a highly innovative approach.
2. The language and illustrations in the paper are clear and effectively convey the characteristics and structure of "Labor Space," making it easy for readers to understand and grasp relevant concepts.
3. The detailed experimental results and analysis in the article convincingly demonstrate the feasibility of pre-trained models in the labor market domain and validate their correctness.
4. The article discusses the integration of pre-trained models with the field of social science and highlights the numerous societal contributions that can result, both from an individual and governmental perspective.

Cons:

1. While the article presents an innovative approach by combining language models with social science, it lacks innovation from a technical implementation perspective. Overall, the choice of BERT as the foundational model, along with fine-tuning using triplet loss, cosine similarity loss, and multiple-negatives-ranking loss functions, comprises the primary approach.
2. The visualization of "Labor Space" in the article lacks sufficient discussion. The paper predominantly focuses on discussing inter-class relationships among entities like firms and industries but does not provide a theoretical explanation of whether their intra-class relationships possess corresponding semantics.

**Questions:**

1. Choosing "tradeable" -> "non-tradeable" as a conceptual axis for discussion is understandable and intuitive. However, could you please clarify why "manufacturing" -> "healthcare" was chosen as the conceptual axis?
2. In the context of "MAPPING HETEROGENEOUS UNITS ON A CONCEPTUAL AXIS," the rationale behind considering the proximity of "real estate," "management," and "public administration" to "healthcare" as reasonable is not clearly explained. Is there a specific basis for this classification?

**Reviewer Confidence:**

3: The reviewer is confident but not certain that the evaluation is correct

**Scope:**

3: The work is somewhat relevant to the Web and to the track, and is of narrow interest to a sub-community

---

### Official Review · Reviewer_5Z6F · 2023-11-27

**Novelty:** 4
**Technical Quality:** 6

**Review:**

The Introduction of the article exhibits a deficiency in references, with numerous statements lacking the necessary support. Notably, the entire Introduction lacks references.

Within the related works section, the authors make several statements without proper references, such as: "Historically, research into this domain has predominantly fallen under two analytical frameworks: (1) understanding the influence of one entity over another, and (2) elucidating the structure of one entity in juxtaposition with another." Where are the references to support these statements?

Furthermore, the related works section lacks references concerning language models applied to the labor market.

There is an absence of detailed descriptions regarding the data utilized. For instance, what information does ONET contain? Is it comprised of job descriptions or curricula?

The article lacks a comprehensive description of the fine-tuning process, including the methodology for selecting hyperparameters.

Moreover, the rationale behind choosing BERT is not adequately explained. Are there alternative choices, and if so, what criteria led to the selection of BERT?

Despite these critiques, the overall results and outcomes of the work are intriguing. The text is well-written, and the figures effectively represent the findings. In summary, the article presents a compelling application of word embedding in an interesting area, although addressing the aforementioned shortcomings would enhance the clarity and credibility of the research.

**Questions:**

1. Could you provide more references in the Introduction section to support the statements made? The lack of unique references raises concerns about the foundation of the introductory content.

2. In the related works section, the article mentions historical research falling under two analytical frameworks. Can you provide specific references to validate these frameworks and strengthen the credibility of the statements?

3. The related works section seems to lack references specifically related to language models applied to the labor market. Could you include relevant literature in this area to enhance the background of your study?

4. The article lacks detailed descriptions of the data used. Can you provide more information about what is included in ONET? Is it primarily composed of job descriptions, curricula, or a combination of both?

5. The fine-tuning process is mentioned, but the article lacks a comprehensive description, including the methodology for selecting hyperparameters. Could you elaborate on the fine-tuning process and provide insights into how hyperparameters were chosen?

6. The rationale behind choosing BERT is not thoroughly explained. Were there alternative models considered, and if so, what criteria led to the selection of BERT over other options?

7. The application of vector arithmetic is interesting. Could you offer additional practical examples to illustrate its use?

**Ethics Review Description:**

-

**Reviewer Confidence:**

3: The reviewer is confident but not certain that the evaluation is correct

**Scope:**

4: The work is relevant to the Web and to the track, and is of broad interest to the community

---

### Decision · Program_Chairs · 2024-01-22

**Decision:**

Accept

**Comment:**

The paper presents an interesting application of large language models to analyse the US labour market. The reviews point out some important limitations and shortcomings of the paper, but they also emphasize its strong points. I think the paper can be accepted, provided that some important corrections are introduced in the final version. In particular, a thorough discussion of the existing literature as discussed, for instance, by reviewer 5Z6F, and a more solid empirical evaluation as requested, for instance, by reviewer f68m